# OpenReview forum: "Verbosity Tradeoffs and the Impact of Scale on the Faithfulness of LLM Self-Explanations"
_ICLR.cc/2026/Conference — Submitted to ICLR 2026_

### Official Review · Reviewer_DkTS · 2025-10-19

**Soundness:** 2
**Presentation:** 3
**Contribution:** 1
**Rating:** 2
**Confidence:** 5

**Summary:**

This paper revisits the Correlational Counterfactual Test (CCT) for measuring LLM self-explanation faithfulness and shows that it can produce misleading results due to verbosity biases in model explanations. The authors systematically analyze this issue and propose phi-CCT to fix these issues. Experiments are conducted across several benchmarks (e-SNLI, ECQA, ComVE) with very numerous models.

**Strengths:**

* S1. The paper is written well, clear, and easy to follow.

* S2. It covers the widest range of LLM models and families I've seen in a faithfulness paper. Unfortunately, this wide variety of models is tested just by one family of faithfulness tests, which is a bummer, see W1.

* S3. The proposed phi-CCT metric is a sensible and well-motivated refinement of CT and CCT, even if the methodological novelty is relatively modest. It is somewhat unfortunate that CCT with its susceptibility to gaming has become an “established” baseline primarily by virtue of publication rather than robustness, necessitating yet another paper to correct its shortcomings.

**Weaknesses:**

* **W1. Narrow focus:** The paper’s evaluation remains narrowly focused on correlational faithfulness metrics (namely CT, CCT and now phi-CCT), without situating results against other faithfulness metrics. Prior work (e.g., [1] and [2]) has shown that different faithfulness metrics can yield strikingly divergent results even on the same models and data, raising the key question of which metric is more reliable, or at least how this new one differs conceptually and empirically. It is somewhat disappointing that the paper remains confined to testing many, many models on just the counterfactual editing family of tests, attempting to refine CCT rather than engaging with the broader concerns of faithfulness evaluation.


* **W2. Improves already flawed tests:** This work improves upon work with fundamental limitations, without addressing or fixing these fundamental limitations. Three concerns:

*Concern 1:* The proposed phi-CCT still inherits conceptual weaknesses from the original CT test, which has been criticized for its lack of correlation with other faithfulness measures [1]. This bears the question of whether CT and its evolutions actually measure the right thing.

*Concern 2:* My extra concern with CT (and all subsequent evolutions of it), is that it works under unreliable assumptions: namely, that if a (random) input change leads to an output change which is not mentioned in the explanation, the model must be unfaithful. Given that neural networks, including LLMs, are inherently sensitive to adversarial perturbations, such an assumption risks labeling all models as unfaithful under some perturbation. The use of random rather than targeted interventions further complicates interpretation: while the authors say they follow Atanasova et al. in applying random edits, this is misleading, because Atanasova et al. explicitly contrasted random and targeted edits and observed important variation between them.

*Concern 3:* CT and evolutions (incl. phi-CCT) need to check whether the edit (the inserted word) was mentioned in the model’s explanation. And they do this check using a simple string-matching approach, which is flawed as it cannot detect synonyms, hypernyms, or negated mentions, and may falsely trigger on irrelevant references (e.g., “this detail is irrelevant”). This triggers the question about the reliability of all CT frameworks.

Overall, it is unclear why continued effort is invested in refining a metric whose foundational premise remains so problematic.

* **W3. No sanity check of the proposed test:** While the paper introduces a new faithfulness measure, it does not include any sanity checks or validation experiments to demonstrate that the metric measures faithfulness. Recent causal evaluation frameworks [2] provide principled ways to test whether a faithfulness metric truly captures faithfulness. Applying such tests would be essential to establish that phi-CCT measures faithfulness rather than surface correlations or artifacts.

References:
 - [1] Parcalabescu and Frank, 2024 cited by the paper
- [2] “A Causal Lens for Evaluating Faithfulness Metrics” (Zaman et al., 2025).
- [3] Atanasova et al. cited by the paper

**Questions:**

see weaknesses, which are all a big question mark about why focus so much on CT?

L 304 "commonmon sense.See A" missing whitespace

---

> ### Author Response · Authors · 2025-11-20
> **Response to DkTS (1/2)**
>
> We thank the reviewer for their comments. We’ve responded below. If there are any remaining concerns which could be addressed by additional analysis or discussion during the review period, please let us know!
>
> **S3.** To clarify in case this was misunderstood, the CCT is *not* susceptible to our notion of “gameability”: see Theorem 2. The CCT did genuinely address this weakness of the CT, which is maximally gameable (Theorem 1). Rather, the limitations we identify are its sensitivity to class imbalance (Figure 3) and verbosity (Figure 4).
>
> **W1.** Thanks for raising this point - this gets to a core issue of the path forward for faithfulness research. However, we believe it *supports* our approach. We agree that inconsistency between faithfulness metrics is an issue: if metrics differ, which, if any, should we trust?
>
> [1] and [2] indeed showed that for their models and datasets, faithfulness metrics gave different results. What would “engaging with the broader concerns of faithfulness evaluation” look like in this setting? Suppose we ran a larger sweep with more metrics, and continued to fail to identify trends - what then? At some point, we need to begin to investigate specific causes of divergence.
>
> This is exactly the aim of our paper: diving into a *specific* cause of inconsistency in faithfulness evaluation and deeply understand it. In particular, explanation verbosity has a major impact on faithfulness as assessed by correlational metrics like the CCT (as well as the CT, as shown in Siegel et al. 2024). We show this mathematically (Figure 3) and empirically (Figure 4). To address this issue, we introduce F-AUROC, which takes into account a model’s performance across multiple faithfulness settings. This indeed appears to reduce noise in evaluation, and allows a clear positive trend to emerge: F-AUROC has a strong monotonic relationship with task performance (Figure 7).
>
> Papers surveying ranges of very different metrics are certainly an important part of the field, and are helpful at identifying issues like lack of consistency. But breadth trades off for depth: [1] evaluated 11 models from 4 families, with at most 2 distinct model sizes per family; and [2] only evaluated 2 different models (qwen-2.5-7b and gemma-2-9b-it). Our larger coverage allows us to identify trends, like those in Figure 7, which could easily be missed with less coverage of model space.
>
> We believe that papers like ours which zoom in on specific causes of inconsistency are an important part of understanding and remedying the current state of faithfulness evaluation.

---

> ### Author Response · Authors · 2025-11-20
> **Response to DkTS (2/2)**
>
> **W2.C1:** Could the reviewer please clarify “the original CT test… has been criticized for its lack of correlation with other faithfulness measures [1]”? [1] assessed the correlation of a number of faithfulness metrics, and indeed found many cases in which metrics disagreed. However, of all the tests, counterfactual edits (the CT) had the *highest* count of positive correlations with CC-SHAP, the authors’ proposed test (Table 5, bar chart).
>
> As mentioned above, [1] also assessed a comparatively limited pool of 11 models from 4 families, with at most 2 distinct model sizes per family. As we’ve found in our work, the faithfulness metrics we consider exhibit quite strong correlations with capabilities; variation means that any small sample of models might miss these trends, but they become clear by considering a wide range of model scales and capabilities (Figure 7).
>
> It’s also worth noting that [1] calls out the CCT as an exception to their general critique: “Because a proper comparison of output tokens requires semantic evaluation, the probability-wise comparison of Siegel et al. (2024) circumvents the evaluation problem.”
>
> **W2.C2:** “if a (random) input change leads to an output change which is not mentioned in the explanation, the model must be unfaithful. Given that neural networks, including LLMs, are inherently sensitive to adversarial perturbations, such an assumption risks labeling all models as unfaithful under some perturbation.” - this misrepresents our claim. We only claim that “If explanations are faithful, we’d expect them to mention impactful interventions more often than non-impactful ones” (L122). We never claim that the existence of adversarial perturbations rules out a model’s faithfulness. But if, under our distribution of interventions, a model fails to mention impactful interventions *more frequently* than non-impactful ones, *this* indicates unfaithfulness. This is one reason we focus on classification-based tests, such as correlation and AUROC, rather than the original CT.
>
> While our interventions are random (following e.g. [1] ([code](https://github.com/Heidelberg-NLP/CC-SHAP/blob/6918e5d73a6ba6716f799dbf2c129f3132e1ef70/faithfulness.py#L319-L324) and (Siegel et al. 2024)[https://arxiv.org/abs/2404.03189]), to avoid the complication of training a dedicated intervention-generating model, as in Atanasova), we do filter sentences for LLM-assessed naturalness, to avoid highly unnatural interventions. See section 4.2.
>
> **W2.C3:** For negative mentions, please see our top-level comment. We also found synonyms and hypernyms to be quite rare; see our examples in Appendix K.
>
> While our initial exploration of LLM judges appeared to show less consistent performance than rule-based matching, as LLMs continue to improve, both phi-CCT and F-AUROC can be measured by using LLM judgments as drop-in replacements for $\mathcal{I}$ and $\mathcal{E}$, and all of our mathematical analysis continues to apply.
>
> **W3.** We did consider [2]. However, note from their limitations section: “For example, we cannot evaluate metrics like Counterfactual Edits (Atanasova et al., 2023), which assess changes in explanations resulting from input modifications. Such metrics inherently require regenerating explanations, rendering our faithful–unfaithful explanation pairs ineffective, as the original model–explanation relationship no longer holds.”

---

> ### Author Response · Authors · 2025-11-27
> **Discussion Period**
>
> Thank you for your initial comments. We hope we have addressed your questions and concerns in our responses last week.
>
> If any concerns remain, we'd be grateful if you could share them so that we have an opportunity for a considered discussion. Alternatively, if we have addressed your concerns, we'd be grateful if you'd consider updating your score.

---

> > ### Comment · Reviewer_DkTS · 2025-11-28
> > **Response**
> >
> > Thanks for the detailed answer. Unfortunately, I am not convinced. This paper doubles down on CT-based testing, which was published with drawbacks, then it needed to come another paper, CCT, to address gameability, and now phi-CCT to address sensitivity to class imbalance and verbosity. It feels like repeatedly patching the same framework. I am not convinced the field needs another paper with a new fix for a new shortcoming of CT when we are in a faithfulness evaluation landscape where different metrics already disagree widely and we haven't addressed that. Now this paper proposes a new metric, but it hasn't been thoroughly compared against the existing alternatives to at least know about the divergences. If missing, are we just setting ourselves up for yet another future paper to make those comparisons?
> >
> > If we could only test that the test measures the right thing -- so, faithfulness? Unfortunately, as the authors point out, a the faithfulness test framework of [4] doesn't apply to CT or CT-derived metrics at all. That's a serious limitation for CT and CT-derived tests because it effectively leaves the metric in an untestable space. I cited [4] because I'm familiar with it and was hoping the authors would find a way to make the faithfulness-test framework applicable here. The standard for new metrics should be testability (to know that this actually measures the right thing), and proposing a non-testable faithfulness metric makes an incomplete paper.

---

> > > ### Author Response · Authors · 2025-12-02
> > >
> > > Thanks for your response.
> > >
> > > > CT-based testing, which was published with drawbacks, then it needed to come another paper, CCT, to address gameability, and now phi-CCT to address sensitivity to class imbalance and verbosity. It feels like repeatedly patching the same framework.
> > >
> > > Is the reviewer arguing that it’s unreasonable for there to be two follow-up papers improving a single method? Wouldn’t this apply to a large fraction of ML research?
> > >
> > > We agree that the original CT was published with drawbacks; hence our current paper. But our interest comes from its valuable core insight: a faithful explanation should *mention things which are impactful*, and we can *determine what is impactful via counterfactual intervention*.
> > >
> > > Furthermore, just because the CT has received more attention in addressing its drawbacks doesn’t imply that other methods are drawback-free. Madsen et al. (2024)’s tests, for example, are subject to trivial gameability (as we discuss in “Response to HRPi (2/2)”), and Matton et al. (2025)’s test uses correlation between prediction impact and explanation importance, which is subject to the sensitivity to verbosity we discuss. We chose to restrict our focus to the CT in order to focus on specific mathematical details and avoid overly complicating the analysis, but we believe this is a step towards addressing these problems more generally.
> > >
> > > > I am not convinced the field needs another paper with a new fix for a new shortcoming of CT when we are in a faithfulness evaluation landscape where different metrics already disagree widely and we haven't addressed that.
> > >
> > > As we discussed in our initial response, our work is a *necessary part* of addressing this situation: we can’t expect methods to agree with *each other* when they may not even agree with *themselves*. And as we discuss above, a major part of disagreement between current metrics may be due to noise caused by factors like sensitivity to class imbalance and verbosity.
> > >
> > > > it effectively leaves the metric in an untestable space.
> > >
> > > Our paper tests our metric by:
> > >
> > > * **Examining qualitative examples.** This is how we can tell our methods are actually measuring faithfulness. See e.g. Table 1: we claim that the second explanation *must be unfaithful*, because the addition of the word “grayish” changes the models’ prediction from neutral to entailment with high probability, but neither explanation gives any mention to color. See Appendix K (Qualitative Examples) for more examples. We can verify faithfulness/unfaithfulness by examining individual examples; the metrics we study are just different ways of aggregating these individual examples into summary statistics.
> > >
> > > * **Comparing with task performance.** This provides empirical evidence that F-AUROC is reducing noise compared to prior methods: Figure 7 shows that as we progress from CT -> CCT -> F-AUROC, the relationship with task performance becomes more and more monotonic, which is consistent with our method reducing evaluation noise due to reducing sensitivity (confirming our mathematical analysis).

---

### Official Review · Reviewer_UnBr · 2025-10-31

**Soundness:** 2
**Presentation:** 3
**Contribution:** 2
**Rating:** 4
**Confidence:** 3

**Summary:**

The paper conducts a large-scale empirical evaluation of LLMs to identify how faithful they are to variations in the prompt through counterfactual testing. They propose a slight variation on existing metrics to account for the full confusion matrix of results and not allow the metric to be trivially gam-able by the LLM being overly verbose in its CoT.

The takeaway message is that as LLMs get bigger, they are more faithful.

**Strengths:**

The empirical evaluation is a primary strength. Analyzing many LLMs and prompt regimes provides a breadth of evidence that is, to my knowledge, the broadest faithfulness study to date. This makes the findings, especially the scaling trends, highly credible.

By demonstrating that phi-CCT is a strong proxy for CCT, it is a nice practical metric for researchers.

**Weaknesses:**

The finding that faithfulness scales with model size is interesting, but I wonder how sound the evaluation really is. Other studies such as Matton et al. [1] have actually found the opposite, and their faithfulness metric was more robust than the author's proposal here in my opinion, as it involved multiple concept edits per instances and an aggregation of this.

I wonder if the proposal to essentially insert a word into the prompt, check for an output mention of it is really measuring the type of faithfulness that is really important, you would have to check if the model is actually saying it is using that word/concept in its decision.

For example, if you insert a word into the prompt, the classification doesn't change, but it mentions the word in such as way as to say that it is ignoring it and not using it in the classification, then I believe this metric would give an incorrect outcome right? Or do I misunderstand?




[1] Matton, K., Ness, R., Guttag, J. and Kiciman, E., Walk the Talk? Measuring the Faithfulness of Large Language Model Explanations. In The Thirteenth International Conference on Learning Representations.

**Questions:**

Can you elaborate on my points about Matton et al. and the potential weakness of your metric because it does not consider if the LLM is actually stating it's using that word/concept in its classification?

I'm also curious why you think your results are contradictory to theirs?

If you can address these points, I will reconsider my score, many thanks.

---

> ### Author Response · Authors · 2025-11-20
> **Response to UnBr**
>
> We thank the reviewer for their thoughtful comments. We’ve responded below. If there are any remaining concerns which could be addressed by additional analysis or discussion during the review period, please let us know!
>
> **“If you insert a word into the prompt, the classification doesn't change, but it mentions the word in such as way as to say that it is ignoring it and not using it in the classification, then I believe this metric would give an incorrect outcome right?”**
>
> We’ve responded to this in our top-level comment.
>
> **“I'm also curious why you think your results are contradictory to theirs?”**
>
> My understanding is that (Matton et al. 2025)[https://arxiv.org/abs/2504.14150] evaluate four models: GPT-3.5, GPT-4o, Claude-3.5-Sonnet (Figure 1), and Llama-3.1-8B (Appendix D.6). The trend appears to be that GPT-4o has the lowest faithfulness (0.56), and Llama-3.1-8B has the highest faithfulness (0.81). Is this the finding the reviewer is referring to?
>
> I found the paper really interesting, and I thought their use of LLMs to generate interventions and assess explanations is a really promising direction. But the major weakness, as acknowledged by the authors in their Limitations section, is the scale of experimentation: “Due to cost constraints, we use a subsample of 30 questions to assess dataset-level faithfulness.”
>
> As such, my understanding is that their overall ranking of model faithfulness is **statistically indistinguishable from null**: Figure 1 shows that their 90% credible intervals for model faithfulness all heavily overlap, and their 90% credible interval for Llama-3.1-8B is [0.49, 1.00], overlapping with the central values of all other models. We believe that a major strength of our paper is the large scale of our experiments, both in terms of our large number of samples enabling narrow CIs, and our evaluation of many different model sizes *within the same families*, isolating differences in capabilities from other factors which might lead to variation between models.
>
> If Matton’s faithfulness ranking represents a real trend, I might speculate that the difference in scaling trend could be due to the domain: as we discuss in section 6, we focus on common sense tasks, where for an accurate model, there is likely no conflict between faithful explanations and human approval. However, on BBQ, if a model’s decision does in fact depend on demographic features like gender or religious identity, a faithful explanation mentioning these features might be *penalized* by a human rater during RLHF, and so more capable models could better learn to strategically avoid faithfulness in situations where they expect to be punished for it.

---

> ### Author Response · Authors · 2025-11-27
> **Discussion Period**
>
> Thank you for your initial comments. We hope we have addressed your questions and concerns in our responses last week.
>
> If any concerns remain, we'd be grateful if you could share them so that we have an opportunity for a considered discussion. Alternatively, if we have addressed your concerns, we'd be grateful if you'd consider updating your score.

---

### Official Review · Reviewer_HRPi · 2025-11-01

**Soundness:** 2
**Presentation:** 3
**Contribution:** 2
**Rating:** 4
**Confidence:** 4

**Summary:**

The goal of the paper is to evaluate the faithfulness of LLM self-explanations. The paper builds on the prior work that proposes the Counterfactual Tests (CT) and Correlational Counterfactual Tests (CCT). The idea of both tests is to insert some tokens in the input and monitor the effect on the output of the model and its explanations. The paper formalizes the two metrics using a common terminology. The paper then uses this formalism to define the notion of alpha gemeability of faithfulness metrics and defines when the metric would be a bad metric. The paper then proposes a new faithfulness test called phi CCT test, which is a simple variation of the CCT test. While the original CCT test depends on probabilities, the proposed test consists of binary indication of the model decision being changed. The metric is then further built upon to measure faithfulness of self-explanations.

**Strengths:**

1. The paper is generally quite well-written and the formalism is easy to follow. For instance, the _C and _D subscript terminology when defining the interventions is very well executed.
2. The framing of the contributed w.r.t the related work is clear. While the key improvement that the paper proposes is a relatively small one (moving from probabilities to discrete labels), it seems quite intuitive.
3. The formal treatment of various metrics is very helpful. It provides clarity on how the various tests relate to each other and also highlight the flaws of CCT for instruction tuned models that respond in natural language. The insight around probabilities not being good indicators in free form generation is also a useful one.
4. The experimental section is large in scale and shows clear, actionable trends in terms of model sizes.

**Weaknesses:**

1. It is not clear from the paper exactly what explainability method is being evaluated and what the implications are. The paper makes some intervention on the inputs by inserting some new words, but that could generate out of distribution sentences that do not mean much in the specific domain while sounding linguistically natural. Did the paper manually characterize the kind of differences the interventions made to the inputs?
2. The paper should also spend some time discussing how these purely computational metrics correlate with human preferences. If the faithful metric value is high, did the humans also find the corresponding explanations useful for some downstream task, e.g., finding bugs in the model behavior, discovering bias, or offering actionable recourse? These are usually the key desiderata behind generating explanations. See for instance [Doshi-Velez and Kim](https://arxiv.org/pdf/1702.08608) and [Wachter et al.](https://arxiv.org/pdf/1711.00399).
3. While the paper is generally well-written, the prompting part in Section 4.1 can use much more detail. For instance, it would be great if the paper discussed the example prompts for prediction and self-explanation and the choices that were made to arrive at them. Were the same prompts used for all models? Could it be that some models could have benefited from prompt tuning?
4. It would also have been great to study the effect of chain-of-thought prompting on the faithfulness of the explanations.
5. In general, it's not clear if the faithfulness test is compatible for all kinds of self-explanations or just the ones considered here? For instance, are feature attribution based, or counterfactual explanations from Madsen et al. are also covered?

**Questions:**

1. How did we extract the model decisions from the free form generations? How effective was this strategy?
2. Line 371: How often did the model follow the instruction on explanation length?

---

> ### Author Response · Authors · 2025-11-20
> **Response to HRPi (1/2)**
>
> We thank the reviewer for their thoughtful questions and comments. We’ve responded below. If there are any remaining concerns which could be addressed by additional analysis or discussion during the review period, please let us know!
>
> **W1.** The explainability method is LLM natural language self-explanations (including CoT). In particular, our method assumes that faithful explanations should be more likely to mention words that had counterfactual impact than words that didn’t.
>
> We did consider the problem of generating out-of-distribution sentences: following [Siegel et al. 2024](https://arxiv.org/abs/2404.03189), we sampled “natural” interventions via rejection sampling with an LLM judge. See section 4.2, and appendix J.2 for our prompt. We chose our threshold of the top 5% most “natural” interventions so that on a large majority of problems, all selected interventions would still make sense. In particular, our judge LLM gives more than 50% likelihood of an intervention making sense for all sampled interventions on 98% of e-SNLI examples, 94% of ECQA, and 92% of ComVE.
>
> From manual inspection, we believe that this method was effective at selecting the most natural interventions, while preserving evaluation on all original dataset examples. See appendix K for randomly selected examples.
>
> **W2.** We agree that traditionally, much work on explainable AI has focused on human preferences. However, our work on faithfulness is motivated by the risk that *true* explanations may *diverge* from human preferences. For example, prior work has identified LLM *sycophancy*: models trained on human preferences are much more likely to agree with a user’s position [Sharma et al. 2023](https://arxiv.org/abs/2310.13548), [Denison et al. 2024](https://arxiv.org/abs/2406.10162). We want to find methods of evaluating whether explanations accurately capture the true factors relevant to model decisions, without the risk of models scoring highly simply by confirming raters’ preexisting beliefs about how a problem “should” be solved.
>
> We believe that a full human study on downstream tasks would be out of scope for our current work, though this would definitely be an interesting and valuable future direction. But our existing empirical results do shine some light on this question: at least for current models and the tasks we study, faithfulness and human preference appear aligned. In particular, models which are larger and more capable (and hence more effective at satisfying human preferences, having been trained at least in part via RLHF) also generally achieve higher faithfulness according to our metrics: see Figure 7, and our discussion in Section 6.
>
> **W3.** Our full prompts are available in Appendix J, “LM Prompts”. We used different prompts for PT vs. IT models, but otherwise prompts were the same between models. Prompts for PT models were taken from Siegel et al. 2024. Additional instructions for IT models were written by us. Our prompt writing process was fairly informal: after writing an initial version of instructions, we checked if any models frequently failed to produce responses with valid formatting (parseable by our regex); if so, we iterated on our prompt and/or regex. After a few rounds of this, most models (other than some of the smallest) reliably produced valid responses. Because prompts include 10-shot examples, we found relatively small impacts from changes to prompt wording.
>
> See also our response to Q2. Please let us know if there are any additional details on prompting that would be helpful.
>
> **W4.** We do study settings where models give their reasons prior to their predictions (explain-then-predict or EP), as well as predict-then-explain (PE). However, we generally failed to find a performance improvement from PE, either in task performance or faithfulness. (This is similar to e.g. [Lanham et al. 2023](https://arxiv.org/abs/2307.13702) (Figure 2), which finds that chain of thought provides substantial gains on math tasks but makes a much smaller difference on other QA domains.) Because of the minor impact for most models, most of our figures presented aggregated results for readability.
>
> We’ve added Figure 13 to our appendix, which explicitly compares between explanation orders.

---

> ### Author Response · Authors · 2025-11-20
> **Response to HRPi (2/2)**
>
> **W5.** Our methods can be applied to a wide range of explanations (both self-explanations and external explanations), including Madsen’s. Our metrics are directly applicable to any test involving:
>
> a. An intervention
> b. A measure of the intervention’s impact on predictions ($\mathcal{I}$)
> C. A measure of whether an explanation mentions the intervened attribute ($\mathcal{E}$)
>
> [Madsens et al. 2024](https://arxiv.org/abs/2401.07927)’s tests *condition their interventions* on the model, in order to effectively achieve $\mathcal{E}=1$ for all interventions. However, this makes their tests subject to the trivial gameability we discuss, similar to the original CT: the CT conditions on $\mathcal{I}=1$, enabling the trivial optimal strategy of ensuring $\mathcal{E}=1$ by repeating the input verbatim; Madsen’s tests condition on $\mathcal{E}=1$, enabling the trivial optimal strategy of ensuring $\mathcal{I}=1$ by e.g. listing or redacting all words in the original explanation. (This was acknowledged as a limitation by the authors: “In principle, the model could redact the entire paragraph, as we don’t constrain the redaction amount.”)
>
> Applying our methods, Phi-CCT and F-AUROC, to Madsen’s tests consists of finding a comparable intervention set with $\mathcal{E}=0$. Rather than simply measuring performance after attribution/redaction, we could compare redaction of words which are identified by the model to words which aren’t, and measure how this is related to whether the model’s prediction changes, either via phi-correlation (Phi-CCT) or AUROC over a range of explanation verbosity thresholds (F-AUROC). A similar approach could be used for their counterfactual explanation test by comparing the model-generated counterfactual edits with a baseline, e.g. edits generated by a model to have comparable edit distance but without the instructions to change the target class.
>
> **Q1.** We match using regular expressions, with the format demonstrated by our few-shot examples. Our full implementation can be found in our supplementary code, corr_faith/experiments/dataset_specific/classification_datasets.py, in the method `parse_prediction`.
>
> Across all models and experiment settings, regular expression parsing fails 10.0% of the time. Failures are mostly driven by small models: for models with at least 10B parameters, parsing only fails 1.1% of the time. Here are the 10 best models, followed by the 10 worst models, in terms of returning responses with valid formatting:
>
> ||Model|Parse Fail Rate|
> |-:|:-|-:|
> |0|openai_api/gpt-4o-mini-2024-07-18|0|
> |1|google/gemma-2-27b-it|3.84615e-06|
> |2|gemini_api/gemini-1.5-pro-002|1.92308e-05|
> |3|gemini_api/gemini-2.0-flash-lite-001|1.92308e-05|
> |4|gemini_api/gemini-1.5-flash-002|4.23077e-05|
> |5|google/gemma-2-2b-it|5.38462e-05|
> |6|gemini_api/gemini-1.5-flash-8b-001|5.38462e-05|
> |7|mistralai/Mistral-Small-24B-Instruct-2501|0.000165385|
> |8|google/gemma-2-9b-it|0.000330769|
> |9|Qwen/Qwen2-72B-Instruct|0.000338462|
> |...|
> |65|google/gemma-7b-it|0.0504308|
> |66|Qwen/Qwen2.5-1.5B-Instruct|0.0692846|
> |67|01-ai/Yi-34B-Chat|0.0839654|
> |68|Qwen/Qwen2.5-0.5B-Instruct|0.186992|
> |69|01-ai/Yi-6B-Chat|0.266846|
> |70|Qwen/Qwen2-1.5B-Instruct|0.406446|
> |71|Qwen/Qwen2-0.5B-Instruct|0.425804|
> |72|google/gemma-2b-it|0.481204|
> |73|Qwen/Qwen1.5-1.8B-Chat|0.837031|
> |74|Qwen/Qwen1.5-0.5B-Chat|1|
>
> **Q2.** Instructions have a significant impact on explanation length. Aggregated over all settings for all instruction-tuned models:
>
> |Instructions|Average Explanation Length (Characters)|
> |:-|-:|
> |Very Concise|118.9|
> |Concise|145.4|
> |Empty|161.1|
> |Comprehensive|281.1|
> |Very Comprehensive|348.8|
>
> To measure how *consistently* models follow instructions, we can compare paired settings: how often is a “Comprehensive” explanation longer than a “Concise” explanation, for the same model, dataset, example, and prompt settings? The following table shows how often <row> yields a longer explanation than <column>:
>
> ||Very Concise|Concise|Empty|Comprehensive|Very Comprehensive|
> |:-|-:|-:|-:|-:|-:|
> |Very Concise|0.50|0.27|0.14|0.06|0.05|
> |Concise|0.73|0.50|0.23|0.08|0.06|
> |Empty|0.86|0.77|0.50|0.13|0.08|
> |Comprehensive|0.94|0.92|0.87|0.50|0.22|
> |Very Comprehensive|0.95|0.94|0.92|0.78|0.50|
>
> For example, a “Comprehensive” explanation is longer than its corresponding “Concise” explanation 92% of the time, showing models follow length instructions fairly consistently.

---

> ### Author Response · Authors · 2025-11-27
> **Discussion Period**
>
> Thank you for your initial comments. We hope we have addressed your questions and concerns in our responses last week.
>
> If any concerns remain, we'd be grateful if you could share them so that we have an opportunity for a considered discussion. Alternatively, if we have addressed your concerns, we'd be grateful if you'd consider updating your score.

---

### Author Response · Authors · 2025-11-20

We thank the reviewer for their thoughtful comments. We’re happy that the reviewers identified our large coverage of LLMs and prompt regimes, our robust and actionable results on scaling trends, and the clarity of our writing and formalism as strengths of our paper. We address specific comments and suggestions in our replies to each reviewer.

We address the following comment that came up in reviews UnBr and DkTS:

**Negative Mentions.** The reviewers ask about cases where the model mentions a word, but specifically says it’s ignoring the word, e.g. “<word> is *not* relevant to my prediction.” We refer to these as “negative mentions”, and discuss them in Limitations. We did check for negative mentions, but found them to be quite rare on our datasets. For example, tables 4, 6, and 8 contain examples where the intervention does not change the model’s predicted class. Of these 30 examples, 11 explanations mention the inserted word, and only one of these (“shoddily”) is arguably a negative mention.

We did briefly explore LLM assessment, as done in Matton et al. 2025, which could potentially allow more semantic evaluation and handle issues like negative mentions automatically. However, when we asked Qwen 2.5 72B-Instruct (our most capable open-weight model) to identify whether explanations mentioned inserted words, we found that it frequently missed words which *did* appear verbatim, or hallucinated mentions even when they didn’t appear. As such, for this work we decided to study the simpler rule-based interventions and mention detection, to allow us to focus on the mathematical formulations of our metrics.

As LLMs continue to improve, however, both phi-CCT and F-AUROC can be measured by using LLM judgments as drop-in replacements for $\mathcal{I}$ and $\mathcal{E}$, and all of our mathematical analysis continues to apply.

---

### Author Response · Authors · 2025-12-02

Unfortunately, the circumstances have prevented reviewers from responding further. We believe we addressed the concerns that arose in the initial reviews, including conducting additional data analysis and visualization, and would have liked the opportunity to confirm this with the reviewers.

We want to highlight one high-level pattern which showed up in the discussion: the tradeoff between breadth vs. depth. In particular, reviewer DkTS listed “Narrow focus” as the first weakness of our paper.

But this focus on depth was intentional, and addresses an issue which is apparent in multiple papers cited by reviewers:

* Reviewer UnBr claimed that “Other studies such as [Matton et al. [1]](https://openreview.net/forum?id=4ub9gpx9xw) have actually found the opposite” of our results; but we identified that due to their small sample size (30 questions), the mentioned result was **statistically indistinguishable from null**.
* Reviewer DkTS cited [Zaman et al., 2025](https://aclanthology.org/2025.emnlp-main.1496/)’s finding that “different faithfulness metrics can yield strikingly divergent results”. However, this paper **evaluated only two models**: qwen2.5-7b and gemma-2-9b-it.

Our paper evaluates 75 different models on a range of prompt regimes and datasets, totalling 2,664 experimental settings with 20,000 counterfactual interventions each (see Appendix I). This makes our results extremely statistically robust, and allows us to confidently identify trends (e.g. due to scaling) which could easily be missed with smaller analyses.

The focus in our field on broad claim scope *to the exclusion of robustness* may be at least partially responsible for the current situation, where results and findings are cited which may have simply been due to chance. We believe it contributes more to the field to make claims which are *true with high confidence* than claims which are *broad but may simply fail to replicate*.

We believe that our work, with its careful mathematical analysis, large-scale experimentation, careful ablation, and openly released code, yields a substantial improvement in signal-to-noise-ratio in terms of understanding faithfulness, and is an important step in remedying this current situation.

---

### Meta-Review · Area_Chair_shpN · 2026-01-11

**Summary:**

The paper conducts an empirical study to measure how faithful LLMs are to variations of a prompt through counterfactual testing. Two of the reviewers were mildly negative and one reviewer was negative, and they all highlighted a number of concerns that prevent me from recommending acceptance, particularly taking into account the exchange of between the authors and the most negative reviewer.

**Reviewer Concerns:**

The concerns regarding CT-based testing are still outstanding.

**Reviewer Scores:**

I do not think the reviewers would have changed their score.

---

### Decision · Program_Chairs · 2026-01-26

Reject